# Development of a methodology for measuring the quality of statutory social workers' complex decision-making

Angela Lilly[1]*, Tim Rakow[2], Jill Manthorpe[3], Benjamin Gardner[4]

1 Department of Psychology, Institute of Psychiatry, Psychology & Neuroscience, King's College London, De Crespigny Park, London, United Kingdom; Department of Psychology, University of Surrey, Guildford, United Kingdom, 2 Department of Psychology, Institute of Psychiatry, Psychology & Neuroscience, King's College London, De Crespigny Park, London, United Kingdom, 3 NIHR Health & Social Care Workforce Research Unit, Faculty of Social Science & Public Policy, King's College London, London, United Kingdom, 4 Department of Psychology, University of Surrey, Guildford, United Kingdom

* angela.1.lilly@kcl.ac.uk; a.lilly@surrey.ac.uk

## Abstract

Registered social workers in English Local Authorities are required to have an expertise in the complex decision-making needed to promote well-being when an adult's own judgement about their well-being and wishes about how to promote it might, in the circumstances, put their well-being at risk. Such circumstances are complex partly because core professional values – promoting autonomy and protecting from harm – can come into conflict. Given the consequential nature of social workers' decisions, it is essential to be able to evaluate the quality of social workers' decision-making. In this paper, we set out the systematic development, in collaboration with expert social workers, of a bespoke methodology to measure decision-making quality and investigate underpinning cognitive processes. Central to our methodology was social workers' consideration of key legal principles. First, we reviewed the research literature to identify existing measurement schemes aspects of which might be suitable for incorporating into our methodology. No existing measurement schemes were found, but we identified a factorial survey vignette-based scheme which seemed promising as the basis for our own methodology. Second, by reviewing statute and case law, we identified 40 key legal principles which social workers should consider in their decision-making. Next, based on these principles, we developed four hypothetical case vignettes to activate decision-making. Finally, we developed four scoring templates, one for each vignette, setting out exemplar judgements and decisions against which practitioners' judgements and decisions could be compared and scored. Our new methodology provides a means of assessing the quality of social workers' decision-making and, as prior- and post-intervention quality can be measured, has the potential to generate evidence of the impact of policy and practice interventions on decision-making.

**Data availability statement:** All relevant data are within the paper and its Supporting Information files.

**Funding:** This project (RF065), including contributions for authors AL and JM, was funded by the School for Social Care Research, a national research school established by the National Institute of Health and Care Research (https://www.nihr.ac.uk/research-funding/social-care). The funders had no role in study design, data collection and analysis, decision to publish, or preparation of the manuscript. The views expressed are those of the authors and not necessarily those of the National Institute of Health and Care Research nor the Department of Health and Social Care.

**Competing interests:** The authors have declared that no competing interests exist.

## Introduction

In England, statutory social workers – that is, those employed by a local authority – occupy just 16% of filled posts in adult social care [1] and, so, best use must be made of their expertise. Statutory social workers in adult social care in England primarily enact the *Care Act* 2014 [2], making judgements and decisions about needs for care and support, such as whether an adult is unable to maintain their nutrition, personal hygiene, home environment or personal relationships, the impact of any such inabilities on their well-being, and the provision of care and support to promote well-being. Social workers employed by the local authority must be registered with the regulatory body, Social Work England. They must comply with the professional standards set by the regulatory body [3] and demonstrate professional capabilities commensurate with their career stage set by the professional body [4], certain of which connote an expertise in decision-making. A registered social worker should be able to, for example, identify, gather and evaluate information from multiple sources; evaluate and integrate this information, which may be conflicting, into judgements about risk in complex circumstances; apply critical reflection and analysis to complex situations; hold different explanations in mind and ensure hypotheses are reviewed to inform decision-making; confidently reason with and apply appropriate legal frameworks and manage conflicting values and ethical dilemmas [3,4]. Decision-making consists of: (1) the identification of potentially valid information; (2) the search for that information; (3) the synthesis of that information into an understanding of and inferential judgements about the situation and (4) a decision, from a number of alternatives, on an appropriate course of action, as explored further in our Methods section. Registered social workers are required to have a high level of expertise in all stages of decision-making, even at an early career stage [4]. Local authorities typically delegate the more *complex* decision-making to registered social workers, in line with national policy and practice guidance [5]. In adult social care, one important type of complex decision-making is the decision-making required when an adult's judgement about their well-being or their wishes about how to promote it might, in the circumstances, put their well-being at risk [6]. Decision-making in these circumstances exhibits established attributes of complex decision-making as defined in the field of decision science, a specialist area of cognitive psychology [7]. That is, there is a high potential for loss for the adult, a high degree of uncertainty about potential outcomes and core professional values are invoked – promoting the adult's autonomy and protecting the adult's well-being from harm – which can come into conflict. Complex decision-making is cognitively demanding and, so, prone to cognitive error and biases, especially when cognitive resources are constrained [8], the detrimental influence of which is well established in such related fields as management decision-making [9], legal decision-making [10,11] and medical decision-making [12,13].

Complex decision-making can be highly consequential. Self-neglect, where an adult neglects their personal hygiene, health or home surroundings or refuses services which might mitigate risk of harm [14] is a prime example of circumstances which, by our definition, require complex decision-making. For example, a person may refuse a meals service because they wish to prepare their meals themself, but

their cognitive decline means that they now cannot do this and, so, they are not eating, with a potentially very adverse impact on their physical and emotional well-being. We do not know the quality of registered social workers' complex decision-making in such circumstances, but we do know that, where cases have been statutorily reviewed because the adult has died and abuse or neglect may have been a factor or the adult has experienced serious abuse or neglect and there are concerns about how they were safeguarded (*Care Act 2014* s.44 [2]), certain recurring themes in decision-making have been highlighted [15,16]. Circumstances in which complex decision-making is required may not always require a safeguarding response but, as safeguarding decision-making is often complex, in line with national policy and practice guidance [5,17], it is typically delegated to social workers, in perhaps 70% of cases [18]. The indications are that the quality of complex decision-making where self-neglect required a safeguarding response would have been improved had there been better formulated judgements about risk, more curious information searches and better balanced legal duties relating to autonomy and protection [19–26] and, by extension, this might also apply to complex decision-making in general. This uncertainty about quality means that there is a pressing need for a methodology to reliably measure the quality of statutory social workers' decision-making in the adult social care setting and explore underpinning cognitive processes. Additionally, such a methodology would help to inform the development of interventions to support decision-making, such as professional supervision and decision support systems, and, because it would be possible to measure decision-making quality prior- and post-intervention, enable the effectiveness of interventions to be assessed.

## Present study: aims and objectives

Our aim was to develop a methodology to measure the quality of statutory social workers' decision-making and investigate underpinning cognitive processes. Our objectives were to develop a methodology to (1) measure the quality of decision-making processes, (2) insofar as legal principles relevant to local authority adult social care appeared to have been considered, and (3) explore underpinning cognitive processes.

## Methodological development

We followed a four-stage development process. In Stage One, we conducted a scoping review to identify any existing methods for measuring the quality of social workers' decision-making that we could draw on in developing our own methodology. In Stage Two, we reviewed relevant statute and case law to extract key legal principles that social workers should consider in their decision-making. In Stage Three, we developed hypothetical case vignettes to activate decision-making processes, in particular the consideration of key legal principles. In Stage Four, we produced four scoring templates, one for each of the vignettes, to enable reliable measurement of decision-making quality. See **Table 1**.

## Ethical approval

Stage One did not require ethical approval. The Social Care Research Ethics Committee gave ethical approval for the design and pilot workshops in Stages Two, Three and Four (approval number 15/IEC08/0037). The Association of Directors of Adult Social Services (ADASS) gave approval for participants to be recruited from four or more local authorities (approval number RG15 024). All prospective participants were provided with written information sheets. Prospective Principal Social Worker participants were additionally provided with verbal and written information about the research at meetings of the London Principal Social Workers Network and those willing to participate were asked to contact the lead author. All participants provided written informed consent.

## Definitions and assumptions

Our methodology development was informed by theory and empirical evidence drawn from the field of decision science, a specialist area of cognitive psychology. We relied on the following definitions: a judgement, which is pre-decisional,

**Table 1. Summary of methodological stages and steps.**

| Stage | Step | Activity | Evidence Sources | Year of Activity |
|---|---|---|---|---|
| Stage One Reviewing the Literature on Decision-Making | Step One Conducting a scoping review | Scoping review | | First iteration 2015 Last updated 2024 |
| Stage Two Identifying Decision-Making Principles | Step One Identifying principles for the social worker role in adult social care | Review of circumstances giving rise to the need for complex decision-making identified in earlier consultation with registered social workers | [27] Law Commission Review of Adult Social Care (Law Com No 326, 2011 [28] | 2014 |
| | Step Two Identifying principles for the local authority adult social care setting | Review of statute law, case law and regulatory body standards | *Care Act* 2014 (Jurisdiction: England) [2] *Mental Capacity Act* 2005 (Jurisdiction: England and Wales) [29] *Human Rights Act* 1988 (Jurisdiction: UK) [30] Associated Provincial Picture Houses Ltd. v Wednesbury Corporation ([1948] 1 KB 223) [31] Council of Civil Service Unions v Minister for the Civil Service [1985] AC 374 [32] Bolam v Friern Hospital Management Committee, [1957] [33] Bolitho v City and Hackney HA, [1997] [34] [3] | 2015 |
| | Step Three Assessing the validity of the principles | Expert practitioner workshops | | 2016 |
| Stage Three Developing Principle-Based Case Vignettes | Step One Developing case vignette structures | Review of legal principles identified in Stage Two, Steps One and Two Review of eligibility criteria for adult social care | *Care Act* 2014 [2] *Care and Support (Eligibility Criteria) Regulations* 2014 (Jurisdiction: England) [35] | 2015 |
| | Step Two Embellishing structures into hypothetical case vignettes | Expert practitioner workshops | | 2016 2017 |
| | Step Three Assessing the validity of the case vignettes | Social worker pilots | | AG and BH vignettes 2016 CI and EK vignettes 2017 |
| Stage Four Developing a Principle-Based Quality Measurement Scale | Step One Developing case vignette scoring templates | Identification of exemplar judgements and decisions | | |
| | Step Two Assessing the validity of the scoring templates | Expert practitioner workshops | | AG and BH vignettes 2016 CI and EK vignettes 2017 |

may be taken as an *"assessment or belief about a given situation based on the available information"* [36] and a decision may be taken as being a *"commitment to a course of action intended to produce a satisfying state of affairs"* [36]. We also drew on the established understanding that decision-making consists of the following stages: (1) the identification of potentially valid information cues to be searched for in the decision environment; (2) the search for that information, including a judgement on when sufficient information has been gathered; (3) the synthesis of that information, often on the basis of a rule, algorithm or some specified strategy into an understanding of the situation and an inferential judgement or series of judgements about it and (4) a decision, from a number of alternatives, on an appropriate course

of action [36]. Synthesising information into judgements about care and support is central to the Care Act 2014 [2] and underpins good quality decision-making [37] and, so, we wanted to develop a methodology which tested all four decision-making stages.

Our methodology development process was also informed by the fact that social workers employed by the local authority are public officers and so are potentially accountable for the quality of their decision-making through the judicial review process. Judicial reviews are not concerned with whether decision outcomes are '*right*', but rather with the decision-making *process*, including whether the process properly conformed to relevant legal principles [38] These include the *reasonableness principle* (Associated Provincial Picture Houses Ltd. v Wednesbury Corporation ([1948] 1 KB 223) [31] and the *properly made principle* (Council of Civil Service Unions v Minister for the Civil Service [1985] AC 374) [32]. The *reasonableness principle* requires public officers to *take into account all relevant factors* and to *take into account no irrelevant factors* in their decision-making. The *properly made principle* requires public officers to make decisions which are not *illegal*, *irrational* nor *procedurally improper*. These requirements display attributes of complex information processing, as defined by decision scientists, which include the need to process large amounts of information from which must be retained only information which is relevant [39–41]. The reasonableness and properly made principles also require public officers' decision-making to be *lawful*, which means that social workers must consider the provisions of the *Care Act* 2014 [2] and associated statute law in their decision-making. Our intention, therefore, was to develop a methodology which would, consistent with the judicial review process, enable the quality of decision-making *processes* to be measured, in particular whether participants appeared to have considered key legal principles.

We also considered it best to develop a methodology in which decision-making was activated by hypothetical case vignettes, which would be better suited to a reliable quality measuring scheme and allow us to avoid ethical challenges which might arise if the quality of decision-making on real cases was measured. We do not see this as problematic, because research indicates that practitioners tend to respond to hypothetical cases and real-life cases similarly [42].

## Stage one: reviewing the literature

### Aim and objectives

Our overall plan was to develop our own methodology for measuring the quality of social workers' decision-making in the statutory adult social care context. We conducted a scoping review to map the research evidence relating to social worker decision-making and identify existing schemes for measuring the quality of social worker decision-making which might be informative for our own methodology. This aim is consistent with the Joanna Briggs Institute's (JBI) definition of scoping reviews [43]: a systematic review was not appropriate as we did not intend to incorporate an appraisal of methodological quality into our review [44].

Our research questions were:

1. What schemes exist for measuring the quality of social workers' decision-making?

2. What methodologies have any such schemes relied on?

Our objectives were therefore:

1. To identify existing schemes for measuring the quality of social workers' decision-making on which we could draw.

2. To evaluate any such schemes for their potential contribution to our own methodology.

The current scoping review was undertaken as part of a broader project to understand statutory social workers' decision-making processes [6].

## Method

Our review method was based on an established five-stage process [45] and structured using appropriate items from the PRISMA Extension for Scoping Reviews [46], building on earlier scoping review iterations [6]. We did not publish a protocol for this review.

### Identifying relevant studies

**Study eligibility criteria.**   Studies were eligible for inclusion where they: (a) related to social worker case decision-making, (b) reported empirical studies, (c) were published in English (d) in or after 1990 and (e) in peer reviewed journals. Criterion (a) was set because relevant methodologies might have been developed for adults' and children's services. Criterion (d) was set because the *Care Act 2014* [2] consolidated existing law and so relevant methodologies might have been developed at any time following the preceding *NHS and Community Care Act 1990* (Jurisdiction: England and Wales) [47].

**Search procedure.**   We used a formal systematic search and filter procedure to conduct a title and abstract search of nine databases, via two platforms (Ovid: Embase, Medline, Global Health, PsychInfo, Social Policy and Practice; ProQuest Social Sciences Premium Collection: Social Science Premium Collection, Social Services Abstracts, Applied Social Sciences Index & Abstracts, Sociology Database). The search string for the Ovid platform search was as follows: ((decision-making or decision making or judg?ment*) and social work*). Fields searched included: abstract or title. Publication type excluded: book; authored book; edited book; chapter; circular; conference abstract; conference paper; correspondence; digital media; dissertation abstract; e-book; editorial; encyclopaedia; erratum; note; online report; review. These publication types were excluded because they did not meet inclusion criteria (b) or (e). We provide search strings for both platforms in Supporting Information (S1). Searches were first run in 2015 but were iteratively updated, with the latest search undertaken on 24th August 2024.

### Selecting relevant studies

**Study selection.**   Screening was undertaken by a single researcher (AL). We first undertook title and abstract screening, conducted simultaneously for efficiency [48]. Full text screening indicated that all studies retained after title and abstract screening were eligible for review.

### Extracting and charting the data

The following data were extracted from all 76 retained papers: sample (adults and/or children); data (type: quantitative and/or qualitative); data collection (self-report: individual and/or group interviews, questionnaires, non-self-report: data extraction from cases; review of case documentation; Q-sort; critical incident; observation or recordings; simulation; case study; secondary data analysis); decision activating material (real cases; hypothetical case vignettes and, if so, vignettes derived from real cases; textbooks or guidance; research team members; social workers or other experts); (subject: decision-making processes and/or factors; decision-making measurement scheme); (findings topic: framing or sensemaking; general information processing; use of knowledge, experience or evidence; processes of synthesising information into inferential judgements; use of heuristics, intuition, emotion; categorisation; legal literacy, legal decision principles or general reference to law).

### Collating and categorising the results

Data from the 76 retained papers were collated using a descriptive-analytical approach, applying a common analytical framework to all papers and collecting standard information on each study [45]. For all studies, data relating to the areas of interest listed were tabulated and subsequently categorised inductively. Data were entered onto a data chart using the

database programme Excel. Although we sought to collect standard data on each paper, it was on some occasions not possible to extract relevant data as it had not been included in the retained paper (Fig 1).

## Results

We identified 76 papers which met the inclusion criteria and were retained for full-text review. We provide summaries of data collected and findings (S2, S3) and references for these in Supporting Information (S4).

### Decision-making quality measurement schemes

None of the 76 retained papers reported existing schemes to measure the quality of social workers' decision-making. Four papers reported schemes in which *aspects* of decision-making were measured, but overall quality was not. Of the four, two papers reported investigations into social workers' accuracy in formulating judgements about the likelihood of significant harm to children [49,50], but accuracy was measured by comparing judgements about future events and outcomes to real-life events and outcomes as they unfolded. One of the four papers purportedly investigated the effect of a care proceedings deadline on quality [51], but quality was measured in terms of decision outcomes, not processes. One of the four papers reported an investigation in which the relationship between information gathering strategies and the generation of

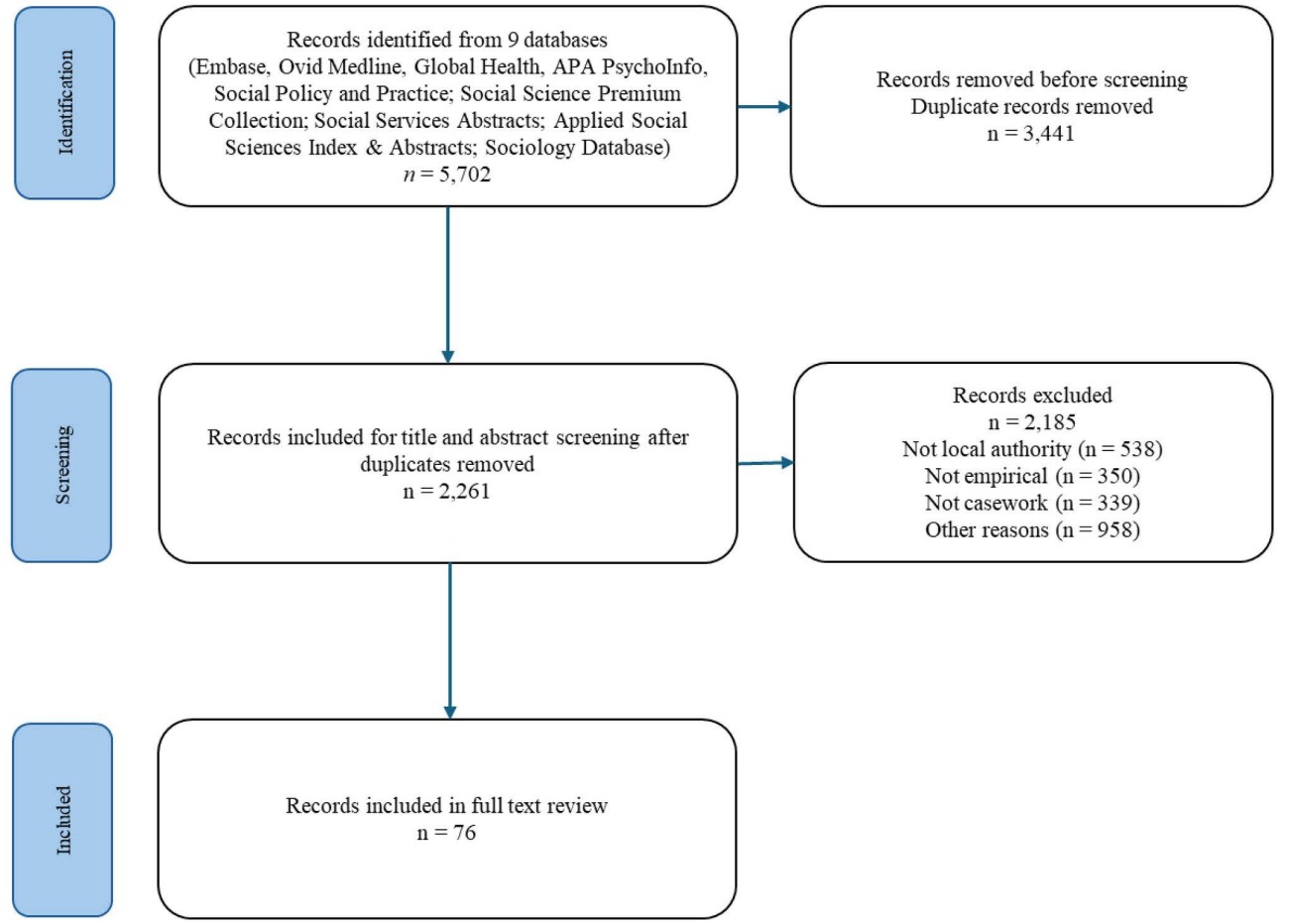

**Fig 1. Prisma flow diagram.**

hypotheses was quantifiably assessed, but not related to decision-making quality [52]. The quality of decision-making was not measured in any of these papers.

Thirty-five papers (46%) reported studies which used one or more non self-report methodologies, either solely or in conjunction with self-report methodologies such as interviews and surveys. The 41 (54%) papers which used self-report methodologies only were not compatible with our objectives due to the subjective nature of these measures. Fifteen papers (20%) reported observations of decision-making, the most common non-self-report method. Decision-making was not quantified in any of these, however, so we could not draw upon these studies in developing our measurement methodology.

## Vignette-based methodologies

Twenty-five papers (33%) reported studies in which hypothetical case vignettes were used to investigate decision-making. Four (5%) reported investigations which relied on case vignettes about adults. In one, case vignettes developed by means of a Q-sort methodology were used to investigate decision-making about financial abuse [53]. In another, a case vignette was used to investigate perceptions of legal intervention to prevent harm despite the adult's wishes [54]. In one further paper, three case vignettes were used to investigate threshold decision-making and evidence based practice [55]. Another paper reported using a vignette-based methodology to activate decision-making about recognising and reporting elder abuse [56]. None of these case vignettes were suitable in themselves for our bespoke methodology. However, one paper reported a vignette development methodology in which randomised permutations of decision-relevant factors, identified by means of a systematic review and a panel of expert professionals, were incorporated into unique factorial survey case vignettes with a set structure and consistent amounts and types of variable information [56]. This seemed a promising approach to developing case vignettes for our measurement methodology.

## Stage one: discussion

We did not identify any measurement schemes that we could incorporate into our methodology in our scoping review. We identified one paper [56] in which a methodology for developing hypothetical case vignettes consisting of randomly generated permutations of decision-relevant factors with a set structure and consistent amounts and types of information was reported which seemed compatible with our aim and objectives. Our scoping review was limited in that, due to resource constraints, data were extracted, summarised and presented by the first author alone. Although scoping reviews should ideally be undertaken by multiple reviewers, most are undertaken by a single researcher [57]. Nevertheless, the findings of our scoping review reinforced the pressing need for a new methodology to measure decision-making quality.

## Stage two: identifying decision-making principles

### Aim and objectives

Our aim in stage two was to identify the key legal principles which statutory social workers in adult social care should consider in their decision-making so that we could incorporate these into our methodology. Our first objective was to identify key legal principles relating to *statutory social workers' decision-making role in adult social care.* Our second objective was to identify key legal principles relating to the *statutory social workers' decision-making as public body officers*. Our third objective was to confirm that the principles identified were key and to identify any which had not been identified in the preceding two steps.

### Step one: principles for the social worker role

To address objective one, we drew on the outcome of an earlier consultation with registered social workers in statutory adult social care led by the first author and a social worker providing expert consultancy to the present research team [27].

The purpose of the consultation, conducted in 2013, was for registered social workers to define their unique role in adult social care in response to the Law Commission Review of Adult Social Care (Law Com No 326, 2011) [28]. The Law Commission Review had been undertaken to provide a clear, modern and effective legal framework for adult social care and its recommendations were implemented in the *Care Act* 2014 [2].

In the earlier consultation, 112 registered social workers from five London local authority adult social care departments attended a total of 11 workshops. Social workers (*n* = 112) were asked to explain their unique role in relation to Recommendation Five of the Law Commission Review, which was that the legal framework for adult social care should include a single overarching well-being principle and that certain factors should be taken into account in deciding how to give effect to the well-being principle. It was generally agreed that social workers' unique role *is to make best use of legal powers to achieve wellbeing for the adult* and that, in deciding how to give effect to the well-being principle, the expertise of registered social workers was specifically required when decision-makers must, as per Law Commission Review No 326, 2011 [28], p. 195–196:

(a) assume that the person is the best judge of their own well-being except in cases where they lack capacity to make the relevant decision;

(b) follow the individual's views, wishes and feelings *wherever practicable and appropriate*;

(c) achieve a balance with the well-being of others, *if this is relevant and practicable*;

(d) safeguard adults *wherever practicable* from abuse and neglect;

(e) use the least restrictive solution where it is necessary to interfere with the individual's rights and freedom of action *wherever that is practicable*.

It was generally agreed that, when these factors applied, decision-making was complex and so required the expertise of registered social workers [27]. Five principles were derived from these five factors.

### Step two: principles for the local authority adult social care setting

To address objective two, a review of relevant legal sources was undertaken by the social worker consultant and the first author. Each legal source was read and debated and 34 legal principles were identified, extracted, summarised and inductively categorised into principles for *public body officer*, *local authority*, *professional* and *registered body* decision-making.

### Principles for public body decision-making

Statutory social workers are public officers and, so, must comply with legal principles relating to public body decision-making. These include the *reasonableness principle*, which stipulates that public officers must, in their decision-making: (a) *be lawful*, (b) *take into account all relevant factors* and (c) *take into account no irrelevant factors* (Associated Provincial Picture Houses Ltd. v Wednesbury Corporation ([1948] 1 KB 223) [31]. These also include the *properly made* principle, which stipulates that public officers' decision-making must be *legal, rational* and *procedurally proper* (Council of Civil Service Unions v Minister for the Civil Service [1985] AC 374) [32]. These principles were reviewed and three key principles derived, including, for example: *the decision must take into account factors that ought to be taken into account.*

### Principles for statutory adult social care decision-making

Statutory social workers in adult social care must, in their decision-making, apply the duties of the *Care Act* 2014 [2] as well as those of associated legislation, such as the *Mental Capacity Act* 2005 [29] and the *Human Rights Act* 1988 [30], and case law. These sources were reviewed and 28 principles derived, including, for example: *if the individual refuses a needs assessment or a service, the social worker must decide whether the individual is experiencing, or is at risk of, abuse or neglect.*

## Principles for professional social worker decision-making

Case law requires that clinical decision-making must be *"in accordance with a responsible body of medical opinion"* (Bolam v Friern Hospital Management Committee, [1957]] [33], which is commonly taken as being relevant *'professional opinion'.* A *"responsible body"* is one whose opinion has a *"logical basis"* (Bolitho v City and Hackney HA, [1997]] [34]. Registered social workers, being a professional body, must, in their decision-making, comply with these case law requirements. This case law was reviewed and two principles derived, including, for example: *the professional decision must be in accordance with a responsible body of professional social work opinion, even if other social workers differ in opinion.*

## Principles for registered social worker decision-making

In England, social workers must meet the specialist standards set out by Social Work England. These standards were reviewed and it was found that all but one were reflected in principles derived from other sources. One standard was not and one principle was derived from this, which was: *if the professional decision is based in part on applied social science, then the social worker must make the decision based on best research evidence as evaluated by them.*

## Step three: assessing the validity of the principles

To address objective three, we held expert social worker workshops with Principal Social Workers. We recruited Principal Social Workers to provide expert professional advice because their role in statutory adult social care is to provide professional leadership and maintain high practice standards. We recruited Principal Social Workers from the London Principal Social Worker Network. The recruitment period started on 25th September 2015 and ran until 9th December 2015, on both of which dates we presented verbal and written information about the research to the London Principal Social Worker Network and asked Principal Social Workers to contact us if they were willing to participate in the expert professional workshops.

The first workshop was held in January 2016 at the offices of the London Association of Directors of Adult Social Services (ADASS). Participating Principal Social Workers were asked to review, evaluate and add to, if necessary, the list of 39 key decision principles identified in steps one and two. We repeated the procedure with the then Chief Social Worker for Adults, England, and an academic lawyer specialising in adult social care law. Participants confirmed that the 39 decision-making principles identified in steps one and two were key for the statutory social worker role in adult social care. They identified one change to an existing principle which they considered necessary: the decision must have regard to the need to protect *people* from abuse and neglect, rather than the *individual* only. The Chief Social Worker confirmed that the 39 principles, as amended, were key. The academic lawyer also confirmed this and added one more principle, derived from the *Mental Capacity Act* 2005 [29] and Code of Practice [58]: *the decision must take into account that any Mental Capacity Act assessment undertaken is decision specific.* We provide these principles in Supporting Information (S5).

## Stage two: discussion

In Stage Two, we identified a total of 40 key principles which statutory social workers in adult social care should consider in their decision-making. These included five principles which related to social workers' unique decision-making role in adult social care and 35 principles which related to social workers' role as a public body decision-maker.

## Stage three: developing principle-based case vignettes

### Aim and objectives

Our aim in Stage Three was to produce hypothetical case vignettes to activate statutory social workers' decision-making, as we had decided that a vignette-based, non-self-report methodology would be most suitable for our measurement

scheme. Our first objective was to develop a standardised structure for the case vignettes to ensure that they incorporated key legal principles and consisted of similar amounts and types of information, drawing on the vignette-based methodology identified in our scoping review [56]. Our second objective was to embellish vignette structures into four realistic hypothetical case vignettes capable of activating social worker decision-making. Our third objective was to confirm that the hypothetical case vignettes were realistic.

## Step one: developing case vignette structures

To address objective one, the key legal principles produced in Stage Two were reviewed by the social worker consultant and the first author to identify a sub-set of principles which would provide the most relevant basis for our vignettes. Additionally, the *Care Act* 2014 [2] and the *Care and Support (Eligibility Criteria) Regulations* 2014 [35] were reviewed to identify and incorporate legal principles relating to eligibility for care and support so that the vignettes would realistically depict eligible adult social care cases. This included a review of provisions relating to well-being in the *Care Act* 2014 [2] as the *Care and Support (Eligibility Criteria) Regulations* 2014 [35] stipulate that an adult's illness or impairment and impact on outcomes must have a significant impact on well-being. Thirty-seven relevant legal principles were identified, extracted, summarised and inductively categorised into six categories, four consisting of eligibility principles and two of intervention principles. We provide these principles in Supporting Information (S6). The four eligibility categories were: (1) areas of social need, conflated from outcome eligibility principles (*Care and Support (Eligibility Criteria) Regulations* 2014 s. 2 (2) (a) – (j)) [35]; (2) illness and impairment eligibility principles (*Care and Support (Eligibility Criteria) Regulations* 2014 s. 2 (1) (a)) [35]; (3) complexity principles relating to social workers' unique decision-making role and (4) areas of well-being principles (*Care Act* 2014 [2] s.1 (2) (a)-(i) [35]. The two intervention categories were: (1) *Care Act* 2014 [2] principles and (2) associated legislation and case law principles (e.g., *Mental Capacity Act* 2005 [29], *Human Rights Act*, 1998 [30]). We decided to base each vignette on eight principles, one from each of the six categories plus two additional '*wild card*' principles drawn from the intervention categories so that each vignette contained sufficient information. Using the "Randbetween" function in Excel, we generated 25 random permutations of sets of eight principles, each of which was suitable as a vignette structure.

## Step two: embellishing structures into case vignettes

To address objective two, we collaborated with expert social workers to embellish vignette structures into realistic hypothetical case vignettes. We did this in three expert professional workshops: two in January 2016, when all four vignettes were produced and a third in March 2017, when the two vignettes yet to be utilised in our studies were reviewed. Participants were Principal Social Workers recruited as outlined above (Stage Two, Step Three). We presented the expert professionals with the 25 principle permutations, asked them to select the most realistic four and, drawing on their professional experience, to embellish these principles into hypothetical case vignettes. Participants substituted principles from the list of 40 produced in Stage Two if they felt this added to their validity, ensuring that principles from each category remained included. Participants paid particular attention to elaborating on the principles related to complex decision-making as defined by the profession, such as that it might not be appropriate to follow the adult's wishes, feelings, views and beliefs, to ensure the vignettes were suitable for registered social workers' expertise. We produced four case vignettes each consisting of between 22 and 32 cue-containing sentences set out on a single side of A4 paper. We provide the case vignettes in Supporting Information (S7). To avoid stereotyping via names, bigrams were used to identify each vignette's main 'character' (AG, BH, CI and EK). For example, CI may be summarised as follows:

> CI is a woman with decreasing mobility due to multiple sclerosis, about whom information has recently come to light to indicate that her husband has been increasingly angry with her, causing her real fear, and has been said to have been seen with his hands around her throat on a couple of occasions. She has denied there is anything wrong.

## Step three: assessing the validity of the case vignettes

To address objective three, we piloted the case vignettes with registered social workers in two workshops. We recruited registered social workers via the London Principal Social Worker Network. The recruitment period for the first pilot workshop was as for the expert practitioner workshops, starting on 25th September 2015 and ending on 9th December 2015, on both of which dates we asked Principal Social Workers to ask registered social workers in their authority to contact us if they wished to participate in the first pilot workshop. The recruitment period for the second pilot workshop effectively ran from October 2016 to May 2017, as we liaised with the then Chief Social Worker and London Principal Social Workers Network ongoingly.

In the first pilot workshop, held on 26th January 2016, we asked social workers (*n* = 8) to read case vignettes AG and BH and write down their free text judgements and decisions about these vignettes in a workbook with instructions we had produced for the purpose. Participants were asked *how typical* of real-world cases the vignettes were, and *how comprehensible* the vignettes and the task instructions were, each using a 7-point Likert scale (1 = strongly disagree, 7 = strongly agree). Responses indicated that the vignettes had external validity *("These are the sort of cases I would normally be asked to make decisions or recommendations on"*: *M* = 6.00) and were comprehensible *("The vignettes were easy to read and understand"*: AG and BH *M* = 6.63). Participants' qualitative feedback was that the AG and BH vignettes and the decision-making task were realistic. We subsequently tested the usability of the workbook in a think aloud procedure where we asked a Principal Social Worker recruited from the London Network to say *'everything that they are thinking…as if they were alone in the room speaking to themselves'* [59]. We produced a final workbook which related to case vignettes AG and BH. We provide this workbook in Supporting Information (S8).

In the second workshop, held on 24th May 2017, we asked social workers (*n* = 9) to read vignettes CI and EK and write down their free-text judgements and decisions in a workbook which we had updated for the purpose. Participants reported that the vignettes had external validity *("These are the sort of cases I would normally be asked to make decisions or recommendations on"*: CI and EK *M* = 5.86) and were comprehensible *("The vignettes were easy to read and understand"*: *M* = 5.71). Participants' qualitative feedback was that the CI and EK vignettes and the decision-making task were realistic. We subsequently revised the workbook to better promote the complexity principles and produced a workbook for case vignettes CI and EK. We provide this workbook in Supporting Information (S9).

## Stage three: discussion

In Stage Three, we produced four hypothetical case vignettes, in collaboration with expert social workers. The case vignettes were each based on randomly generated permutations of principles relating to complex decision-making, where an adult's judgement and wishes might, in the circumstances, put their well-being at risk of harm, as well as principles relating to the *Care Act* 2014 [2] and associated legislation.

## Stage four: developing a principle-based measurement scale

### Aim and objectives

Our aim in Stage Four was to develop a scoring system to allow decision-making quality to be measured. Our first objective was to develop a scoring template for each case vignette, setting out exemplar answers. Our second objective was to assess the validity of the scoring templates and to establish an acceptable degree of tolerance on exemplar answers.

### Step one: developing case vignette scoring templates

To address objective one, the social worker consultant and first author identified judgements and decisions which were considered appropriate because, in making them, relevant key legal principles would likely have been considered in the decision-making process. The key legal principles for each vignette were the eight principles on which each had been based plus additional relevant principles identified by the social worker consultant. Each vignette therefore had a minimum

of at least eight appropriate judgements and eight appropriate decisions. There was not necessarily a simple direct relationship between a judgement or decision and a principle and, so, we decided not to weight any exemplar judgements or decisions. A decision was made to ask participants to make up to five judgements and five decisions on two case vignettes to allow for a sufficiently gradated scale and to allow each participant to make judgements and decisions about two different vignettes. Each judgement and decision would be compared to the answers in the scoring template and given a point if they aligned sufficiently with the exemplar. Scoring for each judgement or decision was binary (the 'point' was either awarded or not) and the sum of these points was treated as an ordinal score, which could range from 0 to 20. A 21-point scale was considered to be sufficiently gradated to test differences between experimental (decision-making intervention) and control (no intervention) conditions.

## Step two: assessing the validity of the case vignette scoring templates

Our second objective was to assess the validity of the scoring templates and to consult expert professionals on an acceptable degree of tolerance on exemplar answers. We held two workshops to do this, one on 6th September 2016, after participant data for AG and BH had been collected, and one on 25th September 2017, after participant data for CI and EK had been collected. The first workshop was attended by three Principal Social Workers recruited from the London Network and was held at the London offices of the Association of Directors of Adult Social Services. The recruitment period for this workshop effectively ran from 20th July 2016, when Study One data collection ended, to 5th September 2016, the day before the first workshop. The second workshop was attended by five Principal Social Workers also recruited from the London Network and was held at King's College London. The recruitment period for this workshop effectively ran from 12th June 2017, when Study Two data collection started, to 24th September 2017, the day before the second workshop. During both periods of time, we provided timely updates, verbal and written, for the then Chief Social Worker and the London Principal Social Workers Network, and asked Principal Social Workers who were willing to participate in the workshops to contact us. Both workshops were attended by the social worker consultant and facilitated by the first author. In both workshops, participating Principal Social Workers were provided with eight participant responses, each consisting of free text judgements and decisions about two case vignettes. In the first workshop, Principal Social Workers applied the scoring templates to score the quality of judgements and decisions. They did this individually and then, in each workshop, arrived at a consensus on the quality scores. In the second workshop, Principal Social Workers did this for four participants' responses and assessed the quality of a further four by, instead, applying their professional judgement so that we could subjectively gauge how consistent the scoring templates were with professional judgement. Expert professionals in both workshops also provided qualitative feedback on the scoring templates and on how similar a free-text answer should be to the judgements and decisions in the scoring template to be given a point. In both workshops, expert professionals were asked to identify any judgements and decisions in the sample of participant data which they considered would also be an appropriate response.

Expert professionals confirmed that the scoring templates enabled the quality of free-text judgements and decisions to be measured and identified no additional exemplar judgements or decisions. Expert professionals stipulated that a free-text response must include sufficient detail: for example, an answer would need to identify the nature of the suspected abuse and the suspected perpetrator and not just state that suspected abuse was a problem. The quality scores arrived at via the experts' professional judgement seemed consistent with the scores arrived at by applying the scoring template, providing further confirmation that the scoring templates were aligned with professional judgement. The scoring templates were therefore approved. We provide the scoring templates as Supporting information (S10).

## Stage four: discussion

In Stage Four, we produced four scoring templates, one for each hypothetical case vignette, collaborating with registered statutory social workers to do so. The scoring templates set out exemplar judgements and decisions against which

free-text judgements and decisions could be compared and, if sufficiently aligned, likely meant that relevant legal principles had been considered, and so could be given a 'point'.

## General discussion

In England, registered social workers are required to have expertise in complex decision-making which may arise when an adult's judgement about their well-being or wishes about how to promote it may, in the circumstances, put their well-being at risk [6]. Social workers' decision-making can be highly consequential in these circumstances. It is essential, therefore, that social workers are provided with interventions which demonstrably support their decision-making. Collaborating with expert professionals, we developed a new methodology to measure the quality of social workers' decision-making, insofar as key legal principles for the adult social care setting are considered, and investigate underpinning cognitive processes. Additionally, as the quality of decision-making can be measured pre- and post- intervention, our methodology will allow the effectiveness of interventions to support decision-making to be evaluated.

Our methodology provides a system for measuring the quality of statutory social workers' decision-making in adult social care. Our final products consist of: a set of 40 key legal principles which statutory social workers in adult social care should consider in their decision-making, four hypothetical case vignettes based on key legal principles to activate decision-making and four scoring templates, one for each vignette, setting out exemplar judgements and decisions against which social workers' judgements and decisions can be compared and, if sufficiently aligned, given points. Importantly, our scheme incorporates key legal principles which social workers should consider in their decision-making, including principles at the crux of decision-making complexity – promoting the adult's autonomous wishes and protecting the adult's well-being from harm.

Strengths of our vignette-based methodology are that practitioners tend to respond to hypothetical cases and real-life cases similarly [42] and that decision-making quality can be compared across vignettes as they include consistent amounts of similar information [56]. However, our methodology is not without potential limitations. First, it could be argued that written case simulations overlook the potential importance of visual and other non-verbal cues [60]. However, there are many real-world occasions when social workers cannot rely on visual cues, such as when they receive information by telephone or in writing or when cases are discussed in supervision. In mitigation, we collaborated with expert professionals to ensure our vignettes typified adults whose circumstances were such that social workers' expertise in decision-making would be required and ratings from a panel of social workers confirmed that the vignettes were seen as realistic. Second, it could be argued that our methodology is based on a simple additive approach to assessing quality which implicitly assigns equal weight to each of the underlying principles and overlooks the possibility that some principles may have a greater influence on quality than others. However, should future work suggest that some principles should be weighted more heavily than others, our scoring system could easily be adapted to reflect this. Third, it could be argued that our methodology relies on an implicit assumption that participants have considered key legal principles in their decision-making if judgements and decisions they have made are given a point. In mitigation, we asked participants to make explicit which legal principles they considered key for each vignette, enabling associations between decision-making quality and legal principles to be tested.

## Conclusion

We have developed what, to the best of our knowledge, is the first reliable methodology for assessing the quality of statutory social workers' decision-making in the adult social care setting insofar as key legal principles appear to have been considered. Our methodology will, firstly, facilitate robust future work to document the quality of social workers' decision-making processes and so better understand their decisions. Secondly, our methodology can be used to assess the impact of policy and practice interventions by permitting measurement of decision-making quality before and after an intervention and so add to the evidence base surrounding interventions designed to support social workers' decision-making.

## Supporting information

**S1. Scoping Review Search Strings.**
(DOCX)

**S2. Supplemental Table One.**
(DOCX)

**S3. Supplemental Table Two.**
(DOCX)

**S4. References for Scoping Review Retained Texts.**
(DOCX)

**S5. Decision-Making Principles.**
(DOCX)

**S6. Case Vignette Decision-Making Principles.**
(DOCX)

**S7. Principle Based Case Vignettes.**
(DOCX)

**S8. Workbook for AG and BH Case Vignettes.**
(DOCX)

**S9. Workbook for CI and EK Case Vignettes.**
(DOCX)

**S10. Case Vignette Scoring Templates.**
(DOCX)

## Acknowledgments

We are grateful to all the social workers who contributed to the development of our methodology. In particular, we thank the social worker who provided expert consultancy to the present research team and we thank the then Chief Social Worker, Principal Social Workers and other social workers who so helpfully collaborated with the research team.

## Author contributions

**Conceptualization:** Angela Lilly, Benjamin Gardner.

**Data curation:** Angela Lilly.

**Formal analysis:** Angela Lilly, Benjamin Gardner.

**Funding acquisition:** Angela Lilly, Jill Manthorpe.

**Investigation:** Angela Lilly.

**Methodology:** Angela Lilly, Jill Manthorpe, Benjamin Gardner.

**Project administration:** Angela Lilly.

**Supervision:** Tim Rakow, Jill Manthorpe, Benjamin Gardner.

**Validation:** Tim Rakow, Jill Manthorpe, Benjamin Gardner.

**Writing – original draft:** Angela Lilly.

**Writing – review & editing:** Tim Rakow, Jill Manthorpe, Benjamin Gardner.

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
