## [Decision Letter · Decision Letter 0]

Dear Dr. Lilly,

Thank you for submitting your manuscript to PLOS ONE. After careful consideration, we feel that it has merit but does not fully meet PLOS ONE’s publication criteria as it currently stands. Therefore, we invite you to submit a revised version of the manuscript that addresses the points raised during the review process.

Both reviewers agreed on the need to better specify you empirical method and participants selection.  

We look forward to receiving your revised manuscript.

Kind regards,

Massimo Finocchiaro Castro, PhD

Academic Editor

PLOS ONE

Journal Requirements:

AL

PO65

School for Social Care, National Institute of Health Research

https://www.sscr.nihr.ac.uk/

No the funders did not have these roles.

Reviewers' comments:

Reviewer's Responses to Questions

**Comments to the Author**

1. Is the manuscript technically sound, and do the data support the conclusions?

Reviewer #1: Yes

Reviewer #2: Yes

2. Has the statistical analysis been performed appropriately and rigorously?

Reviewer #1: N/A

Reviewer #2: Yes

3. Have the authors made all data underlying the findings in their manuscript fully available?

Reviewer #1: Yes

Reviewer #2: Yes

4. Is the manuscript presented in an intelligible fashion and written in standard English?

Reviewer #1: Yes

Reviewer #2: Yes

Reviewer #1: Thank you for the opportunity to review your submission. The paper was well written and reflected a methodologically and ethically rigorous research design. The research question and resulting findings are highly significant to the profession. The knowledge created in this work is new and has the potential to positively influence a number of developments, including professional education, policy and future research.

I have only minor issues for consideration in revision.

The research sets out to develop a measure of decision-making quality. I would like to see some brief comment added to acknowledge that that this measure was based solely on legal principles. The research aims indicate that the study was to develop a methodology to measure quality of decision-making processes 'including' application of legal principles. While these legal principles may incorporate or overlap with other factors in quality (such as taking into account service users' views and preferences) they appear to be the sole basis of the measure, not one which was 'included' in a wider methodology.

References to legislation with nationalities in brackets were not explained and could be confusing. E.g. Care Act 2014 (England) or Human Rights Act 1988 (England) appear to be indicating the geographical extent of this legislation but it is not clear. This could be amended or explained to strengthen this minor element of presentation.

Reviewer #2: This is an important area of research and the authors give a thorough account of their methods in this paper.

I would recommend that the authors state how they are defining complex decision making in the introduction. In lines 88-90, the authors state that, “The profession has defined complex decision-making as arising when an adult’s judgement about their well-being or their wishes about how to promote it put their well-being at risk of harm….”. However, this example is problematic because it reflects one form of complex decision-making in social work, where a range of other complex decisions exist. It is ok to use reference 6 as an example of an important type of complex decision-making, but the authors need to give a clearer definition of this concept, drawing on decision-making literature. A lot of this information is presented in the methods section (lines 163-177), so I would recommend this be sketched out in the introduction with pointers to the methods section.

When reading the introduction, I thought that the paper was focussing specifically on adult safeguarding as the examples given in the introduction all pertain to this. You give safeguarding as an example in lines 88-90 and go onto give examples of self-neglect (also categorised as safeguarding under the Care and Support Statutory guidance) and findings from Safeguarding Adults Reviews (lines 106-122). However, it becomes clear later in the paper that the remit for the paper is broader than this focussing on principles of public body decision making / legal principles. Given this, I think your examples given in the introduction should be broader. This problem recurs in the examples given also pertain to safeguarding whilst the focus of the project is broader. I would also suggest this section is amended, so the scope of the project is better represented (lines 643-644).

**Do you want your identity to be public for this peer review?** For information about this choice, including consent withdrawal, please see our Privacy Policy

Reviewer #1: No

Reviewer #2: No

---

## [Author Response · Author response to Decision Letter 1]

29 Apr 2025

Response to Reviewers

We have reviewed the manuscript to ensure that it meets PLOS One style requirements. We have made some minor revisions to the main body and author formatting to ensure compliance – thank you for the guidance on this.

We have amended the Role of Funder statement as follows (lines 38-43).

Funding. This project, including contributions for authors 1 and 3, was funded by the School for Social Care Research, a national research school established by the National Institute of Health and Care Research. The funders had no role in study design, data collection and analysis, decision to publish, or preparation of the manuscript. The views expressed are those of the authors and not necessarily those of the National Institute of Health and Care Research or the Department of Health and Social Care.

Thank you for changing the online submission form.

We have reviewed the reference list and believe it to be complete and correct, with no retracted papers cited.

4. Reviewer One Comment One

The research sets out to develop a measure of decision-making quality. I would like to see some brief comment added to acknowledge that that this measure was based solely on legal principles. The research aims indicate that the study was to develop a methodology to measure quality of decision-making processes 'including' application of legal principles. While these legal principles may incorporate or overlap with other factors in quality (such as taking into account service users' views and preferences) they appear to be the sole basis of the measure, not one which was 'included' in a wider methodology.

We are grateful for this feedback and have addressed this throughout the manuscript. For example, we have revised the Abstract to include the statement that: “Central to our methodology was social workers’ consideration of key legal principles.” (lines 61-62)

We have revised the Introduction to include the statement that: “Our objectives were to develop a methodology to (1) measure the quality of decision-making processes, (2) insofar as legal principles relevant to local authority adult social care appeared to have been considered in decision-making, and (3) explore underpinning cognitive processes.” (lines 144-145)

We have revised the Discussion to include the statement that: “Collaborating with expert professionals, we developed a new methodology to measure the quality of social workers’ decision-making, insofar as key legal principles for the adult social care setting are considered, and investigate underpinning cognitive processes.” (lines 653-654)

5. Reviewer One Comment Two

References to legislation with nationalities in brackets were not explained and could be confusing. E.g. Care Act 2014 (England) or Human Rights Act 1988 (England) appear to be indicating the geographical extent of this legislation but it is not clear. This could be amended or explained to strengthen this minor element of presentation.

We have addressed this by stating which is the relevant jurisdiction for each Act when first mentioned and thereafter removing the jurisdiction reference.

6. Reviewer Two Comment One

I would recommend that the authors state how they are defining complex decision making in the introduction. In lines 88-90, the authors state that, “The profession has defined complex decision-making as arising when an adult’s judgement about their well-being or their wishes about how to promote it put their well-being at risk of harm….”. However, this example is problematic because it reflects one form of complex decision-making in social work, where a range of other complex decisions exist. It is ok to use reference 6 as an example of an important type of complex decision-making, but the authors need to give a clearer definition of this concept, drawing on decision-making literature. A lot of this information is presented in the methods section (lines 163-177), so I would recommend this be sketched out in the introduction with pointers to the methods section.

We agree and have revised the Introduction to reflect that this is one type of complex decision-making by including the statement that: “In adult social care, one important type of complex decision-making is the decision-making required when an adult’s judgement about their well-being or their wishes about how to promote it put their well-being at risk.” (originally lines 88-90, now lines 100-102).

We have summarised decision-making stages, drawing on decision-making literature, in the Introduction and referred to our expansion on this in our Method section (lines 92-96).

7. Reviewer Two Comment Two

When reading the introduction, I thought that the paper was focussing specifically on adult safeguarding as the examples given in the introduction all pertain to this. You give safeguarding as an example in lines 88-90 and go onto give examples of self-neglect (also categorised as safeguarding under the Care and Support Statutory guidance) and findings from Safeguarding Adults Reviews (lines 106-122). However, it becomes clear later in the paper that the remit for the paper is broader than this focussing on principles of public body decision making / legal principles. Given this, I think your examples given in the introduction should be broader. This problem recurs in the examples given also pertain to safeguarding whilst the focus of the project is broader. I would also suggest this section is amended, so the scope of the project is better represented (lines 643-644).

We are grateful for this advice and have revised the Introduction by considerably reducing the narrative on safeguarding adults reviews and making clear that, whilst complex decision-making may arise in safeguarding situations, complex decision-making is not at all limited to safeguarding. We have referred to ways in which safeguarding adults reviews might shed relevant light on the quality of complex decision-making, such as better balancing of legal duties, but have moved away from a detailed description of safeguarding adults reviews. We have then referred, as per the original manuscript, to the pressing need for a methodology to measure the quality of decision-making (now lines 113-132).

We have also referred to “complex decision-making which may arise when an adult’s judgement about their well-being or wishes about how to promote it may, in the circumstances, put their well-being at risk.”, removing the word harm here and in one or two other places, to emphasise that complex decision-making is broader than safeguarding (line 650).

---

## [Editor Report · Decision Letter 1]

Development of a methodology for measuring the quality of statutory social workers’ complex decision-making

PONE-D-25-02726R1

Dear Dr. Lilly,

We’re pleased to inform you that your manuscript has been judged scientifically suitable for publication and will be formally accepted for publication once it meets all outstanding technical requirements.

Kind regards,

Massimo Finocchiaro Castro, PhD

Academic Editor

PLOS ONE
---

## [Editor Report · Acceptance letter]

PONE-D-25-02726R1

PLOS ONE

Dear Dr. Lilly,

I'm pleased to inform you that your manuscript has been deemed suitable for publication in PLOS ONE. Congratulations! Your manuscript is now being handed over to our production team.

Kind regards,

on behalf of

Prof. Massimo Finocchiaro Castro

Academic Editor

PLOS ONE